# Tackling Neonatal Sepsis—Can It Be Predicted?

**DOI:** 10.3390/ijerph20043644

**Published:** 2023-02-18

**Authors:** Špela But, Brigita Celar, Petja Fister

**Affiliations:** 1Faculty of Medicine, University of Ljubljana, 1000 Ljubljana, Slovenia; 2Department of Paediatric Intensive Care, Division of Paediatrics, University Medical Centre Ljubljana, 1000 Ljubljana, Slovenia

**Keywords:** neonate, sepsis, infection, diagnostic markers of infection, C-reactive protein, procalcitonin, prediction of sepsis, application for predicting neonatal sepsis

## Abstract

(1) Background: Early signs of sepsis in a neonate are often subtle and non-specific, the clinical course rapid and fulminant. The aim of our research was to analyse diagnostic markers for neonatal sepsis and build an application which could calculate its probability. (2) Methods: A retrospective clinical study was conducted on 497 neonates treated at the Clinical Department of Neonatology of the University Children’s Hospital in Ljubljana from 2007 to 2021. The neonates with a diagnosis of sepsis were separated based on their blood cultures, clinical and laboratory markers. The influence of perinatal factors was also observed. We trained several machine-learning models for prognosticating neonatal sepsis and used the best-performing model in our application. (3) Results: Thirteen features showed highest diagnostic importance: serum concentrations of C-reactive protein and procalcitonin, age of onset, immature neutrophil and lymphocyte percentages, leukocyte and thrombocyte counts, birth weight, gestational age, 5-min Apgar score, gender, toxic changes in neutrophils, and childbirth delivery. The created online application predicts the probability of sepsis by combining the data values of these features. (4) Conclusions: Our application combines thirteen most significant features for neonatal sepsis development and predicts the probability of sepsis in a neonate.

## 1. Introduction

Sepsis is a potentially life-threatening syndrome of organ dysfunction induced by a dysregulated response to an infection [1]. The risk of developing sepsis is especially high in neonates due to their vulnerable and underdeveloped immune system [2]. We differentiate between neonatal early-onset sepsis (EOS), which develops in the first 72 h of life, and late-onset sepsis (LOS), which develops after that. In general, EOS is most commonly caused by Group B Streptococcus (GBS) and *Escherichia coli* (*E. coli*) [3,4], whereas the primary agents of LOS appear to be Coagulase-negative Staphylococci (CoNS) and *Staphylococcus aureus* (*S. aureus*), followed by *E. coli* and GBS [3,5]. The onset is highly dependent on perinatal factors such as gender, gestational age, birth weight, Apgar score, type of childbirth delivery, and maternal factors [2,6]. Early signs of sepsis are often subtle and non-specific [6], ranging from temperature instability (hypothermia more often than fever) to respiratory and cardiac clinical signs, poor feeding, and lethargy [5]. Yet, the clinical course of neonatal sepsis can be contrastingly rapid and fulminant [6]. Clinical presentation is due to its variability and heterogeneity always interpreted with regard to the non-specific markers of infection, most commonly the C-Reactive Protein (CRP) and Procalcitonin (PCT), whereas blood cultures remain the gold standard for diagnosing sepsis, even with the delay in results [2]. Over the years, researchers have focused on finding a single diagnostic test or a collection of tests that would help neonatologists reach precise and fast recognition of a neonate with sepsis. Furthermore, it would help with early diagnostics, followed by specific and effective treatment [7]. The focus was predominantly on serum concentrations of CRP and PCT, the complete blood cell count (CBC), and the ratio between immature and total neutrophil count (I:T) [8]. For the early detection of neonatal sepsis, several researchers focused on combining various markers of the disease and building prediction models out of them. In 2022, Sahu et al. reviewed multiple studies on forecasting algorithms for neonatal sepsis and recognised predictors among maternal, neonatal, and laboratory parameters. In addition, a different array of promising factors was put forward for the early detection of EOS, LOS, and neonatal sepsis in general. The reviewed prediction models were mostly built by logistic regression (in 7 out of 10 studies) [9]. However, other statistic methodologies could also prove suitable for forecasting algorithms. In 2021, Neamtu et al. recognised the decision-tree approach as being useful for predicting the outcomes of neonatal brain injury. They pointed out that the decision-tree approach has also been put to test for forecasting other aetiologies, including neonatal infections [10]. Regardless of the applied algorithm, all researchers concluded that through further developments in this field, prediction modelling could have large implications for early recognition of adverse outcomes of neonates in the future [9,10]. To this day, tackling neonatal sepsis remains a topical issue for paediatricians all over the world. The incidence of neonatal sepsis on a global level is estimated at 2824 neonates per 100,000 live births, or 2.8%, largely determined by the income level of the country [11]. While its high morbidity and mortality rates are still the primary concern of the less developed countries, the developed countries swing between the antibiotic over-prescription and under-prescription on account of acting too late. With respect to that, we conducted research to obtain insight into the burden of neonatal sepsis in Slovenia and to analyse diagnostic markers used in our population. Our aim was to propose an approach that could more easily and precisely differentiate neonates with sepsis from those without sepsis. We attempted to build an application that could calculate the probability of sepsis and could hence aid clinicians in a timelier diagnosis and well-reasoned care of neonates.

## 2. Materials and Methods

A retrospective clinical study was conducted on neonates who were treated at the Clinical Department of Neonatology of the University Children’s Hospital in Ljubljana from 2007 to 2021. In total, we collected data for 998 infants, but half (501) of them had missing data and were excluded. Out of the remaining 497 infants, 403 neonates met the inclusion criteria of being diagnosed with neonatal sepsis based on the International Classification of Diseases (P36.0–P36.9), and 94 represented the control group. These were the neonates treated due to other, non-infectious diseases. Patient data were collected from a paediatric computer information system called ISPEK. Data collected consisted of perinatal factors, clinical signs, and laboratory markers of infection. Observed perinatal factors were gestational age, birth weight, head circumference, gender, type of childbirth delivery, 5-min Apgar score, maternal risk factors (e.g., prolonged rupture of membranes, intrapartum antimicrobial prophylaxis, prenatal exposure to steroids), presence of a meconium-stained amniotic fluid, and the age of onset. Out of clinical signs, we observed changes in body temperature (>38.5 °C/<36.0 °C), the presence of tachycardia (>160 beats/minute), tachypnoea (>60 breaths/minute), inappetence, prolonged capillary refill (>3 s), irritability, lethargy, and jaundice. Among laboratory markers of infection, serum CRP and PCT concentrations and CBC were observed. The neonates with a diagnosis of sepsis were separated based on their blood culture results, clinical signs, and laboratory markers of infection and were later divided into three groups: proven, probable, and suspected sepsis group (Table 1). For the purpose of more accurate model training, the neonates in the proven and probable groups were defined as septic, and the neonates in the suspected sepsis and control group were defined as non-septic.

Using machine learning, we developed several models in Python for predicting neonatal sepsis. We used the best-performing model as the core of our application. Out of all the observed features, we only used seventeen with the least missing values. Since missing values in our dataset had no hidden dependency on any other variable, the type of missing data was Missing Completely at Random (MCAR). To approximate them, we applied the Nearest Neighbour imputation, which is intuitive and yields good results when there are not many features and observations [12,13]. The dataset was split into training and testing sets; the training set contained 397 neonates (80%), and the testing set contained 100 (20%). The distribution of septic and non-septic neonates was equal in both sets. The training set was used to train the following models: logistic regression, decision tree classifier, Support Vector Machine, K-Nearest Neighbours, and Random Forest. We evaluated the models’ efficacy with standard classification metrics: classification accuracy, sensitivity, specificity, positive and negative predictive value, Area under Curve, and F1 score. The most promising model was Random Forest (Classification and Regression Trees or CART); the others were discarded. The Random Forest algorithm comprises multiple separate decision trees. A figure of one of the decision trees used in Random Forest is in Appendix A (Figure A1). We improved the accuracy of the Random Forest model by training it on a modified dataset, which contained only the thirteen most important features. These features were serum concentration of CRP, the age of onset, immature neutrophil percentage, serum concentration of PCT, lymphocyte percentage, leukocyte count, thrombocyte count, birth weight, gestational age, 5-min Apgar score, gender, presence of toxic changes in neutrophils, and the type of childbirth delivery. Feature selection was based on feature importance metrics. In our case, we applied feature selection methods that are part of the Random Forest algorithm. Furthermore, for fine-tuning the hyperparameters, grid search with k-fold Cross-Validation was used. We used five folds to get the best parameters. After we got the best parameters, we trained the model again with ten-fold Cross-Validation.

## 3. Results

In total, 497 neonates were included in the analysis. The percentage of neonates in the sepsis group (proven and probable sepsis) was proportionate to the number of neonates in the non-sepsis group (suspected sepsis and control group), which is 49% and 51%, respectively (Figure 1). 

### 3.1. Pathogens

The pathogens, isolated from the investigated blood cultures, were Gram-positive (G+) and Gram-negative (G-) bacteria. Neonatal sepsis was more often caused by G+ bacteria (in 52 neonates, or 57%) and was predominantly late onset (in 80 neonates, or 88%). The most common pathogens were *E. coli* and GBS, which were isolated in 27% and 25% of all blood cultures, respectively, followed by CoNS in 15% and *S. aureus* in 13%. The distribution of causative agents is shown in Figure 2. We isolated *S. epidermidis* and *S. haemolyticus* among CoNS, *E. faecalis* and *E. faecium* from *Enterococcus* spp., *K. oxytoca* and *K. pneumoniae* from *Klebsiella* spp., *E. cloacae* from *Enterobacter* spp., and placed *Serratia marscensens* and *Acinetobacter junii* among other G- bacteria. 

### 3.2. Perinatal Factors

Neonates included in our study did not differ much in the general perinatal characteristics, such as gestational age, birth weight (mostly term), and head circumference (Table 2). They were mostly male and were more often delivered by vaginal birth. Proven sepsis was most commonly of late onset. Though the median 5-Minute Apgar score was nine in all groups, it can be seen from the interquartile range that the neonates in the sepsis group were more likely to require immediate medical attention after birth than those in the non-sepsis one. Based on our findings, a meconium-stained amniotic fluid or maternal factors did not act as important risk factors for developing neonatal sepsis. 

### 3.3. Clinical Presentation of Neonatal Sepsis

Neonates with sepsis did not exhibit a specific clinical presentation, nor were the clinical signs of infection often present. Even though almost all the investigated clinical signs were more commonly observed in septic neonates, they were still not very frequent, as only one reached above 50% (Table 3). By definition, the neonates in the control group did not have any clinical signs of infection.

### 3.4. Laboratory Markers of Infection

The median leukocyte count did not differ much in septic and non-septic neonates, but it can be seen from the interquartile range that septic neonates had lower leukocyte counts. Neonates with sepsis had a higher percentage of immature and mature neutrophils, as well as higher I:M and I:T ratios, in addition to lower lymphocyte percentage. Thrombocyte count was highest in the control group. Toxic changes in neutrophils were frequently present in the sepsis group. The levels of serum CRP and PCT concentrations were higher in septic neonates (Table 4).

### 3.5. The Model for Predicting Neonatal Sepsis

Table 5 shows standard classification metrics for all trained models.

Based on classification metrics, the best-performing model was Random Forest (Table 6). This model was later embedded into an online application.

### 3.6. Feature Selection for Model Training

Figure 3 shows thirteen out of seventeen features that had the highest importance values based on feature selection methods. They were selected for model training. The best-performing model was used for our application, which allows the user to enter the patient’s data of those features and calculates the probability of sepsis in the neonate. The application can be accessed at https://sepsa.blang.eu/ (accessed on 26 December 2022).

## 4. Discussion

The main goal of this research was to determine whether neonatal sepsis can somehow be quantified, and therefore, predicted, even in the early stages of its development. Firstly, we established which perinatal factors act as risk factors for developing sepsis, which clinical signs, if any, are the most indicative of the disease, and which diagnostic markers of infection are both the most common and most significant for reaching the diagnosis. We focused on data that are easily obtained and observed and could act as a significant as well as an early marker of neonatal sepsis. Secondly, we classified features based on their importance for predicting neonatal sepsis and embedded a selected combination into an online application. The inclusion criteria were commonness and feature importance or significance in neonatal sepsis prediction. In addition, the number of selected features had to be just right to not cause overfitting of the prediction model. After entering the patient’s data into all the required fields, the application calculates the probability of sepsis in that specific neonate. The result is a percentage, and a higher percentage corresponds to a higher chance of a neonate being septic.

Based on our findings, five perinatal risk factors for neonatal sepsis were included in our model. Those were male gender, vaginal delivery, lower 5-min Apgar score (<7), lower gestational age, and low birth weight (LBW) (Figure 3). The male gender appeared to be more vulnerable to developing neonatal sepsis (Table 2), which was put forward by other researchers as well [3,4]. However, there were more male neonates in the non-sepsis group as well, which could imply that in general, males are of greater susceptibility in the days following their birth, compared to females. Furthermore, the risk of EOS was greater if the mother was colonised with GBS, which is recognized as the primary pathogen responsible for EOS [4]. Thus, neonates delivered vaginally are generally at higher risk of developing sepsis due to vertical (mother-neonate) transmission of pathogens [14]. A lower Apgar score indicates that the neonate needs immediate medical attention after birth and can be, to some extent, associated with a greater risk of infection [2,15,16,17,18]. Preterm and LBW neonates are more susceptible to infections due to their less mature immune system and the deficit of protective maternal IgG antibodies that cross the placenta of term neonates [3,6]. In our study, LOS was more common than EOS (the median onset time was 9.5 days). On one hand, the reason could lie behind efficient antenatal screening for GBS; on the other hand, the neonates were most often admitted from their homes (out-of-hospital setting), which could contribute to a higher level of LOS cases. To continue, none of the observed clinical signs turned out as reliable indicators for sepsis, which is in line with the findings of other researchers. The clinical presentation is often referred to as non-specific, as an ill-looking neonate can be potentially septic. However, it could also have meningitis, pneumonia, or suffer from prematurity or asphyxia-related complications [19]. Hence, the clinical presentation was not included in the application. Nevertheless, we recognize that in our research, the lower reliability of clinical presentation might be related to the overall lower frequency of observed signs of infection in neonates (Table 3). This could be due to the error of subjective assessment and insufficient anamnestic data of parents whose neonates developed sepsis outside the hospital. Out of all the laboratory values tested, only the counts of the leukocytes and thrombocytes, the percentages of immature neutrophils and lymphocytes, the presence of toxic changes in neutrophils, and the serum concentrations of CRP and PCT acted as significant for diagnosing neonatal sepsis (Figure 3). In a neonate, CBC must be interpreted carefully concerning the clinical signs and other laboratory markers of infection. Contrary to older children and adults, the normal range of the total white blood cell count in a neonate is quite wide (9–30 × 10^9^/L); hence, it does not act as a precise tool for diagnosing an infection [20,21]. In our study, leukocyte count showed an importance of only 0.06 (Figure 3). The leukocyte count in a neonate is generally on the high-value side on the first day of life and decreases later [22]. Based on the literature, a low leukocyte count is associated with EOS as well as LOS, and a high leukocyte count is associated only with LOS [23,24]. While interpreting leukocyte count, we should take into consideration that the neutrophil count tends to be lower at lower gestational age and that other clinical conditions can also affect it (perinatal asphyxia, maternal fever, meconium aspiration syndrome, and delivery route) [22]. Out of all haematological features included in model training, immature neutrophil percentage showed the highest importance value (0.11) (Figure 3). The medians of immature neutrophil percentages and I:T ratios were clearly higher in septic neonates (Table 4). The I:T ratio may be, based on the literature, the most sensitive indicator of neonatal sepsis out of all haematological indices [22]. In addition, during an immune response to an infection, changes in lymphocyte count also occur [25]. Lymphocyte count showed an importance value of 0.08 with its median lower in septic neonates (Table 4). However, we can mention that nowadays, the neutrophil/lymphocyte ratio is becoming a more promising inflammation marker than lymphopenia. It has a significant positive correlation with EOS and will surely be the subject of future studies [25]. Furthermore, certain toxic changes in leukocytes, such as cytoplasmic vacuolization and toxic granulations, can occur as a result of an infection. These changes are generally considered late markers of infection and are never present in healthy neonates [26,27,28]. In our study, they were more frequently present in the septic group of neonates (Table 4). Besides changes in leukocyte count and toxic changes in leukocytes, thrombocytopenia also appeared to be related to increased odds of infection in our research. This finding is similar to other studies [23,24]. To continue, CRP is an acute-phase protein and the most commonly used marker of infection in neonates. It is synthesised in the liver in response to infection or tissue damage [26,29]. In our study, the serum concentration of CRP acted as the most important feature for diagnosing sepsis with the importance value of 0.37 (Figure 3). Its values were much higher in the sepsis group in comparison to the non-sepsis group, where the median was 0 (Table 4). Despite that, we must be careful at interpreting its concentration in the first days after birth because of different antenatal and perinatal factors (e.g., prolonged rupture of membranes, intrapartum antimicrobial prophylaxis, prenatal exposure to steroids), which can also affect its values [30]. In addition to that, meconium aspiration syndrome, surgical procedures, chorioamnionitis, and perinatal anoxia can also cause higher serum CRP concentration [31,32,33]. With that in mind, the neonates with other valid reasons for high values of CRP, such as severe birth asphyxia, viral infections and severe brain haemorrhage, were excluded from our study. Moreover, PCT also acts as an acute-phase protein whose serum concentration increases with the gravity of the infection. Contrary to the serum concentration of CRP, local bacterial infections, and viral and other non-bacterial infections result in a lower increase in the serum concentration of PCT [20,34,35,36]. While interpreting laboratory results in the first 24 h after birth, we have to keep in mind a physiological increase in serum PCT concentration [30]. Based on our findings, the median serum PCT concentration was higher in the sepsis group and around 0 in the non-sepsis group, similar to serum CRP concentration (Table 4). The importance of serum PCT concentration was, on the other hand, lower (0.1) (Figure 3). Around 20% of the values of serum PCT concentration were missing and we had to apply imputation techniques to approximate them, which could have had some impact on the results. Nevertheless, serum PCT concentration is, based on the literature, more relevant for diagnosing the early stages of neonatal sepsis than serum CRP concentration [37]. It would be interesting to compare our importance values with a similar, potentially prospective study, which would include more measurements of serum PCT, as it is being more regularly monitored nowadays.

In comparison, Kaiser Permanente developed a Neonatal Early-Onset Sepsis Calculator which is easily accessible online [38]. It has been recognized as a convenient tool that significantly impacts patient care and reduces antibiotic administration, the number of laboratory tests, and admissions to the neonatal unit without increasing mortality or readmissions of the neonates [39,40]. In this calculator, the probability of neonatal sepsis is calculated based on the values of perinatal risk factors. The probability is given as the risk of EOS per 1000 births, which can also be modified considering the neonate’s clinical presentation. There are several strengths of the calculator. Firstly, the required data are usually quite easily accessible, unlike the results of the laboratory markers, which take several hours. Secondly, after calculating the risk of developing neonatal sepsis, the calculator also recommends further management of a neonate. Thirdly, Kaiser’s solution of minimising the impact of a neonate’s immaturity due to preterm birth was setting the bar at 34 gestation weeks, which could be taken into consideration when making further improvements to our application. On the other hand, Kaiser’s calculator only predicts the risk of developing EOS, and not EOS and LOS combined, as it is in our case. The advantage of including all septic neonates, regardless of the onset time of the disease, is the broader use of such an application. However, we do recognize that higher applicability could come on the account of a lower precision of the calculated probability of sepsis. In light of our work, Kaiser’s calculator also lacks the values of laboratory markers of infection such as the serum concentrations of CRP and PCT, and CBC. Their reasoning behind this omission is that the serum concentration of CRP has poor positive predictive value and that it has not been proven to be useful for diagnosing sepsis in asymptomatic neonates. In addition, observing CBC is only recommended as an additional test and the serum concentration of PCT is not mentioned at all. We could argue that, though serum concentration of CRP is in fact a late marker of infection, it still adds value to the calculation of neonatal sepsis prediction, especially when considered alongside other laboratory markers of infection. In our study, the serum concentration of CRP proved to be the most important feature associated with sepsis, though its importance value was still only 0.37. However, if we add the importance values of serum PCT concentration and the percentage of immature neutrophils, which appeared as the most important features (Figure 3) besides serum CRP concentration, the end value of feature importance is 0.58. Furthermore, the value increases if we add even more features, which are significant for diagnosing neonatal sepsis; however, adding too much could be a double-edged sword leading to overfitting of the model. Finding the balance and just the right number of features were our goals when creating the application. Nevertheless, we do acknowledge the effect of the predetermined cut-off values of the laboratory markers and that their adjustment or finer tuning could potentially lead to more accurate results. Lastly, in the pursuit of predicting neonatal sepsis as precisely as possible, the two calculators could perhaps be united; the number of required features would thus be expanded or even reduced. 

The main limitation of our research is that the study was retrospective. That allowed us to get quite a big sample of almost a thousand neonates; however, only approximately half of them were included in the final analysis due to missing values. Moreover, the included neonates had missing data as well, which was the reason for using the Nearest Neighbour imputation. Even though imputation is an established approach in machine learning, we acknowledge that it can increase variance and cause overfitting. It is also worth mentioning that obtaining a sufficient amount of blood for analysis purposes can be challenging due to the neonates’ lower total blood volume. In addition, as every neonate needs to be treated individually and the selection of medical tests chosen rationally, the selection of laboratory markers of infection amongst the patients differed. While the serum concentration of CRP was examined almost routinely, it was the serum concentration of PCT that was most frequently lacking. In addition, since the study was retrospective, we could not determine the significance of newly proposed biomarkers for sepsis, such as presepsin [2]. Another limitation of our research is not collecting data of blood urea nitrogen (BUN) and creatinine. Based on recent developments, BUN is strongly associated with the presence and severity of neonatal sepsis [41]. Monitoring creatinine could be useful because septic neonates are at higher risk of developing renal dysfunction and acute kidney injury [42]. However, it is our belief that all this could be field for further research. Should the findings be relevant for prognosticating neonatal sepsis, they could be integrated into our application. Further studies could also collect data on term and preterm neonates separately and determine how maturity and immaturity shape the neonates’ response to an infection. 

## 5. Conclusions

To conclude, the ideal diagnostic marker of neonatal sepsis is still not known. In practice, a diagnosis of sepsis in neonates is reached by combining perinatal risk factors, clinical signs and laboratory markers of infection, and the results of blood cultures. Gathering all the necessary markers can be time-consuming, but an early management of a potentially septic neonate is contrastingly crucial. In light of aiding clinicians in reaching a timely and more precise diagnosis of neonatal sepsis, we created an application that predicts its probability. The application can hopefully prove useful to neonatologists when handling potentially septic neonates and impact patient care altogether. We believe our work shows a promising step towards prediction modelling in the field of neonatal sepsis; however, extensive research and further developments are still much needed. 

## Figures and Tables

**Figure 1 ijerph-20-03644-f001:**
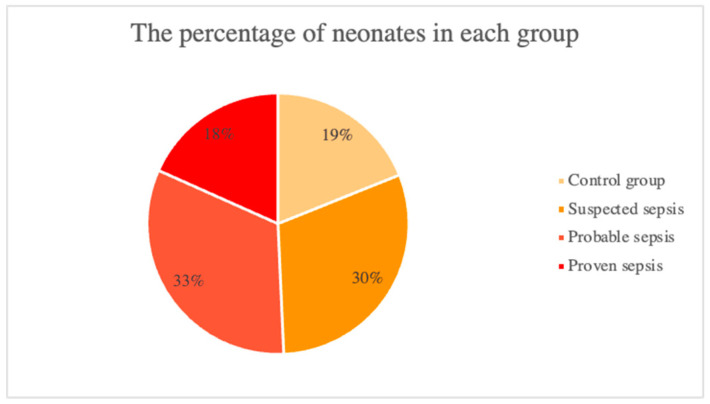
The study groups of 497 neonates: the control group and suspected sepsis comprising the non-sepsis group and the probable and proven sepsis comprising the sepsis group.

**Figure 2 ijerph-20-03644-f002:**
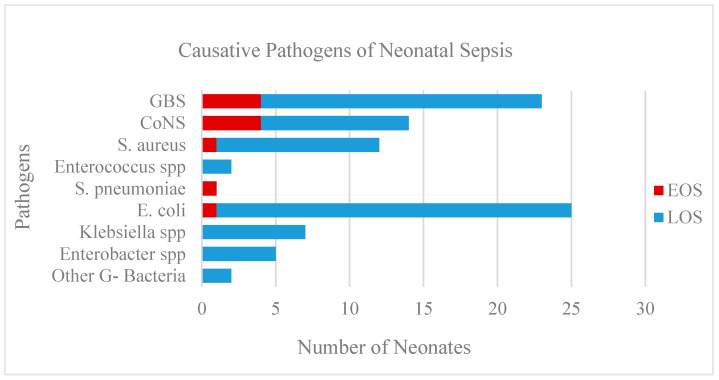
Pathogens isolated from 91 positive blood cultures in the proven sepsis group.

**Figure 3 ijerph-20-03644-f003:**
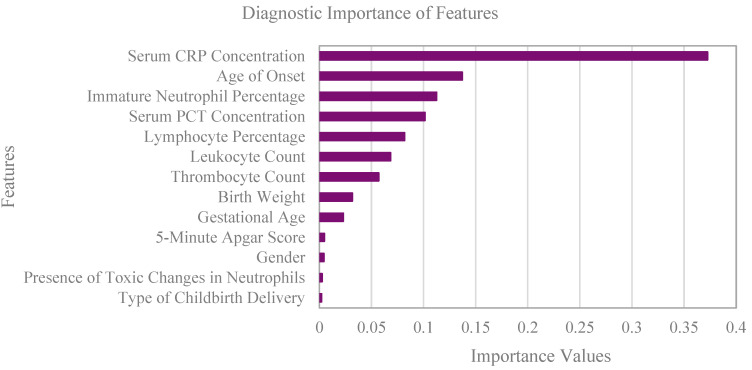
Thirteen features with the highest diagnostic importance, based on which our best-performing Random Forest model, were trained.

**Table 1 ijerph-20-03644-t001:** Separation of neonates diagnosed with neonatal sepsis based on blood culture results, clinical signs, and laboratory markers of infection.

	Proven Sepsis	Probable Sepsis	Suspected Sepsis
Blood culture results	+	-	-
Clinical signs of infection	+	+	uncharacteristic
Laboratory markers of infection	+	+	uncharacteristic

**Table 2 ijerph-20-03644-t002:** Perinatal risk factors for neonatal sepsis and the age of onset.

Perinatal Risk Factors	Proven Sepsis	N’/N (%)	Probable Sepsis	N’/N (%)	Suspected Sepsis	N’/N (%)	Control Group	N’/N (%)
Gestational Age, Me (IQR) [gestation weeks]	39 (36–40)	90/91 (99)	39 (38–40)	160/161 (99)	38 (37–40)	149/151 (99)	39 (38–40)	94/94 (100)
Birth Weight, Me (IQR) [g]	3200 (2460–3645)	91/91 (100)	3400 (2910–3780)	161/161 (100)	3170 (2743–3525)	151/151 (100)	3290 (3010–3628)	94/94 (100)
Head Circumference, Me (IQR) [cm]	34 (32–36)	88/91 (97)	35 (33–36)	156/161 (97)	34 (33–35)	149/151 (99)	34 (33–35)	92/94 (98)
Male gender, n/N (%) *	63/91 (69)	63/91 (69)	103/161 (64)	103/161 (64)	86/151 (57)	86/151 (57)	57/94 (61)	57/94 (61)
Vaginal Delivery, n/N’ (%)	59/89 (66)	59/91 (65)	128/158 (81)	128/161 (80)	97/145 (67)	97/151 (64)	69/94 (73)	69/94 (73)
Apgar 5, Me (range)	9 (2–10)	82/91 (90)	9 (2–10)	152/161 (94)	9 (3–10)	138/151 (91)	9 (5–10)	85/94 (90)
Meconium-Stained Amniotic Fluid, n/N’ (%)	10/65 (15)	65/91 (71)	23/114 (20)	114/161 (71)	24/104 (23)	104/151 (69)	2/84 (2)	10/94 (11)
Maternal Risk Factors (ATB/Steroids/PROM), n/N’ (%)	14/52 (27)	52/91 (57)	21/107 (20)	107/161 (66)	26/68 (38)	68/151 (45)	11/70 (16)	70/94 (74)
Age of Onset, Me (IQR) [days]	14 (9–21)	91/91 (100)	11 (4–19)	161/161 (100)	2 (1–13)	151/151 (100)	7 (3–16)	94/94 (100)

Me = Median. IQR = Interquartile range. n = Variable. N = The number of all data in each of the groups. N’ = The number of all available data in each of the groups (smaller than N due to missing values). Maternal risk factors (ATB/Steroids/PROM): A mother who was treated with antibiotics or steroids prior, during, or after delivery, or had a prolonged rupture of membranes. * n = N’.

**Table 3 ijerph-20-03644-t003:** Clinical signs of infection *.

Clinical Presentation	Proven Sepsis	Probable Sepsis	Suspected Sepsis
Temperature > 38.5 or < 36.0, n/N (%) [°C]	25/91 (27)	35/161 (22)	8/151 (5)
Heart Rate > 160, n/N (%) [beats/min]	43/91 (47)	52/161 (32)	27/151 (18)
Respiration Rate > 60, n/N (%) [breaths/min]	33/91 (36)	66/161 (41)	39/151 (26)
Inappetence, n/N (%)	40/91 (44)	75/161 (47)	35/151 (23)
Capillary Refill > 3, n/N (%) [s]	26/91 (29)	38/161 (24)	8/151 (5)
Irritability, n/N (%)	42/91 (46)	86/161 (53)	37/151 (25)
Lethargy, n/N (%)	21/91 (23)	58/161 (36)	28/151 (19)
Jaundice, n/N (%)	15/91 (16)	43/161 (27)	24/151 (16)

n = Variable. N = The number of all data in each of the groups. * There was no missing data in any of the groups for clinical signs of infection.

**Table 4 ijerph-20-03644-t004:** Laboratory markers of infection.

Laboratory Markers	Proven Sepsis	N’/N (%)	Probable Sepsis	N’/N (%)	Suspected Sepsis	N’/N (%)	Control Group	N’/N (%)
Leukocyte Count, Me (IQR) [×10^9^/L]	11 (6.3–16.8)	91/91 (100)	13.2 (8.0–18.7)	150/151 (99)	12.9 (9.0–18.8)	160/161 (99)	10.4 (8.8–11.9)	94/94 (100)
Immature Neutrophils, Me (IQR) [%]	8 (3–14)	87/91 (96)	4 (1–10)	151/161 (94)	1 (0–5)	132/151 (87)	0 (0–2)	94/94 (100)
Mature Neutrophils, Me (IQR) [%]	56 (42–68)	87/91 (96)	57 (44–68)	155/161 (96)	54 (39–64)	136/151 (90)	34 (26–46)	94/94 (100)
Lymphocytes, Me (IQR) [%]	23 (11–31)	87/91 (96)	22 (17–33)	155/161 (96)	29 (20–42)	136/151 (90)	47 (34–56)	94/94 (100)
I:M, Me (IQR)	0.13 (0.05–0.29)	87/91 (96)	0.09 (0.01–0.19)	151/161 (94)	0.02 (0.00–0.07)	132/151 (87)	0.00 (0.00–0.06)	94/94 (100)
I:T, Me (IQR)	0.11 (0.05–0.22)	87/91 (96)	0.07 (0.00–0.15)	151/161 (94)	0.02 (0.00–0.07)	132/151 (87)	0.00 (0.00–0.06)	94/94 (100)
Toxic Changes in Neutrophils, n/N’ (%)	13/90 (14)	90/91 (99)	20/154 (13)	154/161 (96)	10/133 (8)	143/151 (95)	7/90 (8)	90/94 (96)
Thrombocyte Count, Me (IQR) [×10^3^/L]	272 (191–356)	90/91 (99)	296 (210–402)	158/161 (98)	276 (215–360)	149/151 (99)	322 (259–376)	94/94 (100)
Serum CRP concentration, Me (IQR) [mg/L]	29 (12–79)	91/91 (100)	34 (15–61)	161/161 (100)	0 (0–10)	151/151 (100)	0 (0–0)	94/94 (100)
Serum PCT concentration, Me (IQR) [μg/L]	5.6 (1.5–27.9)	91/91 (100)	1.9 (0.5–10.0)	161/161 (100)	0.4 (0.2–1.0)	114/151 (75)	0.1 (0.1–0.2)	32/94 (34)

Me = Median. IQR = Interquartile range. n = Variable. N = The number of all data in each of the groups. N’ = The number of all available data in each of the groups (smaller than N due to missing values). I:M = The ratio between immature and mature neutrophils. I:T = The ratio between immature and total (mature and immature) neutrophils.

**Table 5 ijerph-20-03644-t005:** Classification metrics for all trained models.

	CA (%)	Se (%)	Sp (%)	PPV (%)	NPV (%)	F1 (%)	AUC (%)
Random Forest	83	81	87	86	79	84	84
Logistic Regression	81	76	86	86	74	80	81
Decision Tree Classifier	82	80	84	84	80	82	82
Support Vector Machine	79	75	83	83	75	78	79
K-Nearest Neighbours	62	35	78	78	35	49	63

CA = Classification Accuracy. Se = Sensitivity. Sp = Specificity. PPV = Positive Predictive Value. NPV = Negative Predictive Value. F1 = F1 score. AUC = Area under Curve.

**Table 6 ijerph-20-03644-t006:** Classification metrics of the best performing Random Forest model after fine-tuning.

CA (%)	Se (%)	Sp (%)	PPV (%)	NPV (%)	F1 (%)
86	82	89	89	82	85

CA = Classification Accuracy. Se = Sensitivity. Sp = Specificity. PPV = Positive Predictive Value. NPV = Negative Predictive Value. F1 = F1 score.

## Data Availability

The data presented in this study are available on request from the corresponding author.

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
