# Peer review of "Tackling Neonatal Sepsis—Can It Be Predicted?"

_ijerph, 2023, doi:10.3390/ijerph20043644_

Round 1
Reviewer 1 Report
Dr. Špela But and colleagues analyzed diagnostic markers for neonatal sepsis and built an application that could calculate its probability. Their approach envisaged “ a retrospective clinical study conducted on 497 neonates who were treated at the Clinical Department of Neonatology of the University Children’s Hospital in Ljubljana from 2007 to 2021. The neonates with a diagnosis of sepsis were separated based on their blood cultures and clinical and laboratory markers; also, the influence of perinatal factors was observed. They trained several machine-learning models for prognosticating neonatal sepsis and used the best-performing model in our application. (3) Results: Thirteen features showed the highest diagnostic importance: serum concentrations of C-reactive protein and procalcitonin, age of onset, immature neutrophil and lymphocyte percentages, leukocyte and thrombocyte counts, birth weight, gestational age, 5-minute Apgar score, gender, toxic granulations, and childbirth delivery. The created online application predicted the probability of sepsis by combining the data values of the aforementioned features.”
The research design and the developed application are very interesting and highly important for clinicians, however, some major issues should be addressed regarding the manuscript.
1. Introduction
Predicting neonatal sepsis based on specific algorithms is definitely of paramount importance. Nevertheless, there is no reference regarding other reports tackling forecasting algorithms, just a short review of the most referred biomarkers. You need to add another paragraph in the introduction in this sense. You could use, for example, the following titles or any other research papers as references :
1. Sahu, P., Raj Stanly, E.A., Simon Lewis, L.E. et al. Prediction modeling in the early detection of neonatal sepsis. World J Pediatr 18, 160–175 (2022). https://doi.org/10.1007/s12519-021-00505-1
2. NeamÈ›u, B.M.; Visa, G.; Maniu, I.; Ognean, M.L.; Pérez-Elvira, R.; Dragomir, A.; Agudo, M.; Șofariu, C.R.; Gheonea, M.; Pitic, A.; Brad, R.; Matei, C.; Teodoru, M.; Băcilă, C. A Decision-Tree Approach to Assist in Forecasting the Outcomes of the Neonatal Brain Injury. Int. J. Environ. Res. Public Health 2021, 18, 4807. https://doi.org/10.3390/ijerph18094807.
2. Materials and Methods
1. Why did the authors not use as inputs in their model also BUN and creatinine, given the recent reports in this respect?. For example Li et al. 2021 demonstrated that BUN was independently linked with the presence and severity of neonatal sepsis. Li X, Li T, Wang J, Dong G, Zhang M, Xu Z, Hu Y, Xie B, Yang J, Wang Y. Higher blood urea nitrogen level is independently linked with the presence and severity of neonatal sepsis. Ann Med. 2021 Dec;53(1):2192-2198. DOI: 10.1080/07853890.2021.2004317.
This should be mentioned as a study limitation in the discussion section
2. Which type of decision trees did you use? This should be clearly presented.
3. What type of software was used to implement the machine learning algorithms? R, Python, SPSS modeller?
4. You should avoid confusion as k-fold Cross-validation for k=10 is ten fold Cross-validation and not ten k-fold Cross-validation
3. Results
A figure with the selected model from the Random-Forest output should be provided and explained based on the mentioned predictors
In the Appendix, you need to present a synoptic table with the total number of cases, including missing data for all the parameters that you mention in the manuscript
Table 5- you should give the performances of all algorithms that have been tested, not just for Random Forest.
Materials and Methods and Discussions
4. You should also mention what type of missingness you got for your dataset. Was it MCAR or MAR or nonignorable? Likewise, you have to document with references from the literature why you chose the Nearest Neighbour imputation for your case.
Author Response
Dear Ms Vinnie He,
Thank you for the opportunity to resubmit our manuscript. We would like to thank you, the editors and the reviewers for their valuable remarks. We have revised the text, taking into account all of the comments and suggestions, and we hope that you will find our revised manuscript suitable for publication.

Reviewer 2 Report
This is an interesting paper that sheds light on the importance of calculations in order to evaluate the risk of infections.
The factors that were taken into account when calculating the sepsis risk are clearly stated but I am not convinced that they are in accordance with the scientifical data known today. For example, is hard for me to believe that CRP is more specific for sepsis than Procalcitonon. It is well known that high values of CRP can appear even in the absence of sepsis.
Also, I don t believe that the APGAR score at 5 minutes can be correlated to the risk of infection because it is calculated using some criteria that are not relevant for infections at only 5 minutes of life. More than that you stated that most of your patients were term neonates with LOS - so that type of patient usually has high APGAR scores.
In our hospital, LOS is very scarce and usually, it s considered an alarm for the lack of hospital cleanliness or a late reaction of the doctors to the neonate suffering. So, again, it is very difficult to believe that you have so many LOS cases, a far more numerous series than EOS cases.
Another question mark is raised by the fact that you stated the clinical signs usually are absent in the presence of sepsis. It contradicts the very basis of neonatology - it is known, for example, that no more than 40% of the blood cultures are positive, but the clinical signs are much more frequent in neonatal sepsis.
For the reasons mentioned above, I consider you should do more thorough research in the literature regarding neonatal sepsis, and do a critical major revision of this paper.
Author Response

(The authors gave the same response as above.)

Round 2
Reviewer 1 Report
The authors substantially improved the quality and soundness of the manuscript, properly addressing all my concerns, and they should be commended for their thoroughness. The manuscript is very interesting and should be published. I have no further comments.
Reviewer 2 Report
There is considerable improvement in this paper. I am glad that you approached all the items indicated by me.
I am sure that in your future articles you will be more careful in the way you present the work behind the text.
Thank you!